# The Multifaceted Role of Cofilin in Neurodegeneration and Stroke: Insights into Pathogenesis and Targeting as a Therapy

**DOI:** 10.3390/cells13020188

**Published:** 2024-01-18

**Authors:** Faheem Shehjar, Daniyah A. Almarghalani, Reetika Mahajan, Syed A.-M. Hasan, Zahoor A. Shah

**Affiliations:** 1Department of Medicinal and Biological Chemistry, College of Pharmacy and Pharmaceutical Sciences, Toledo, OH 43614, USA; faheem.shehjar@utoledo.edu (F.S.); reetika.mahajan@utoledo.edu (R.M.); 2Stroke Research Unit, Department of Pharmacology and Toxicology, College of Pharmacy, Taif University, P.O. Box 11099, Taif 21944, Saudi Arabia; almarghalani@tu.edu.sa; 3Department of Pharmacology and Experimental Therapeutics, College of Pharmacy and Pharmaceutical Sciences, University of Toledo, Toledo, OH 43614, USA; shasan6@rockets.utoledo.edu

**Keywords:** cofilin, neuroinflammation, neurodegenerative diseases, cofilin inhibition, cofilin signaling

## Abstract

This comprehensive review explores the complex role of cofilin, an actin-binding protein, across various neurodegenerative diseases (Alzheimer’s, Parkinson’s, schizophrenia, amyotrophic lateral sclerosis (ALS), Huntington’s) and stroke. Cofilin is an essential protein in cytoskeletal dynamics, and any dysregulation could lead to potentially serious complications. Cofilin’s involvement is underscored by its impact on pathological hallmarks like Aβ plaques and α-synuclein aggregates, triggering synaptic dysfunction, dendritic spine loss, and impaired neuronal plasticity, leading to cognitive decline. In Parkinson’s disease, cofilin collaborates with α-synuclein, exacerbating neurotoxicity and impairing mitochondrial and axonal function. ALS and frontotemporal dementia showcase cofilin’s association with genetic factors like C9ORF72, affecting actin dynamics and contributing to neurotoxicity. Huntington’s disease brings cofilin into focus by impairing microglial migration and influencing synaptic plasticity through AMPA receptor regulation. Alzheimer’s, Parkinson’s, and schizophrenia exhibit 14-3-3 proteins in cofilin dysregulation as a shared pathological mechanism. In the case of stroke, cofilin takes center stage, mediating neurotoxicity and neuronal cell death. Notably, there is a potential overlap in the pathologies and involvement of cofilin in various diseases. In this context, referencing cofilin dysfunction could provide valuable insights into the common pathologies associated with the aforementioned conditions. Moreover, this review explores promising therapeutic interventions, including cofilin inhibitors and gene therapy, demonstrating efficacy in preclinical models. Challenges in inhibitor development, brain delivery, tissue/cell specificity, and long-term safety are acknowledged, emphasizing the need for precision drug therapy. The call to action involves collaborative research, biomarker identification, and advancing translational efforts. Cofilin emerges as a pivotal player, offering potential as a therapeutic target. However, unraveling its complexities requires concerted multidisciplinary efforts for nuanced and effective interventions across the intricate landscape of neurodegenerative diseases and stroke, presenting a hopeful avenue for improved patient care.

## 1. Introduction

Neurodegenerative diseases represent an elusive and growing global health challenge, imposing an ever-increasing burden on individuals, families, and healthcare systems [1]. These diseases, characterized by the progressive degeneration and loss of neurons, manifest in a multitude of forms, including Alzheimer’s disease (AD), Parkinson’s disease (PD), Huntington’s disease (HD), and stroke [2]. Despite their distinct etiologies, these conditions share common underlying cellular and molecular mechanisms contributing to neuronal dysfunction and demise [3]. In this era of intensive neurodegeneration research, cofilin protein has emerged as one of the essential players in the intricate web of neuronal homeostasis and dysfunction [4]. Cofilin, a highly conserved actin-binding protein, has garnered significant attention for its pivotal role in modulating cytoskeletal dynamics, cell motility, and synaptic plasticity [5,6]. Beyond its fundamental cellular functions, mounting evidence has revealed that cofilin has been implicated in neurodegenerative disease pathogenesis, opening up new possibilities for understanding and potentially intervening in the disease [7,8,9].

The focus of this review is to comprehensively explore the multifaceted role of cofilin in neurodegeneration, with a specific emphasis on its implications in stroke pathogenesis. Stroke, a leading cause of morbidity and mortality worldwide [10], is characterized by the sudden interruption of blood flow to the brain, resulting in neuronal injury and inflammation [11]. While significant progress has been made in stroke research, the molecular complexity underlying stroke-induced neurodegeneration and neuroinflammation remains enigmatic [12]. Herein, we embark on a journey through the complicated world of cofilin signaling, aiming to shed light on its involvement in neurodegeneration and its potential as a therapeutic target for stroke and beyond. In the following sections, we will delve into cofilin’s structural and functional aspects, elucidate its role in neurodegenerative diseases, particularly stroke, explore its connection to neuroinflammation, and assess the therapeutic strategies revolving around cofilin inhibition. Additionally, we will discuss the broader implications of targeting cofilin in various neurodegenerative diseases and outline the challenges and future directions in this burgeoning field. As we navigate through this multidimensional landscape, it becomes evident that cofilin holds promise as a critical player in neurodegeneration and as a beacon of hope for innovative therapeutic interventions. In an era where neurodegenerative diseases continue to impose a profound socio-economic burden, understanding the interplay of cofilin within the brain may unlock new avenues for treatment and bring us closer to the ultimate goal of mitigating the devastating impact of these disorders.

## 2. Cofilin: Structure and Function

### 2.1. Molecular Structure and Regulation of Cofilin

Cofilin, a vital actin-binding protein, is ubiquitously expressed in eukaryotic cells and plays a fundamental role in regulating cytoskeletal dynamics [13]. It comes under a group of proteins collectively known as the actin-depolymerizing factor (ADF)/cofilin family, encompassing ADF, also recognized as destrin, along with cofilin-1, the predominant isoform found ubiquitously in non-muscle tissues, and cofilin-2, the primary isoform in mature muscle tissues [14,15] (Figure 1). For this review, our focus will be on cofilin-1, referred to henceforth as cofilin. Its molecular structure and intricate regulatory mechanisms are vital to understanding its multifaceted functions within the cell [16]. Cofilin is a small protein consisting of approximately 166 amino acids. It is highly conserved across species, underscoring its essential role in cellular processes. Cofilin’s secondary structure consists of alpha helices and beta strands. These elements fold and arrange themselves to form specific structural motifs. The secondary structure contributes to the protein’s overall stability and three-dimensional architecture [17]. At the tertiary level, cofilin adopts a well-defined structure. It comprises a central core region enriched in alpha-helical structures surrounded by beta strands. This compact and organized tertiary structure enables cofilin to interact with actin filaments with precision [18].

The primary structure of cofilin contains key functional domains, including the N-terminal actin-binding domain and the C-terminal regulatory domain [19,20]. The N-terminal domain of cofilin is responsible for binding to filamentous actin (F-actin) [21]. Cofilin binds preferentially to ADP-bound actin monomers, leading to conformational changes in F-actin and promoting filament severing and depolymerization [22].
Figure 1NMR structure of human cofilin isoforms [23,24,25]. (https://doi.org/10.2210/pdb1Q8G/pdb; https://doi.org/10.2210/pdb7M0G/pdb, (accessed on 2 January 2024)).
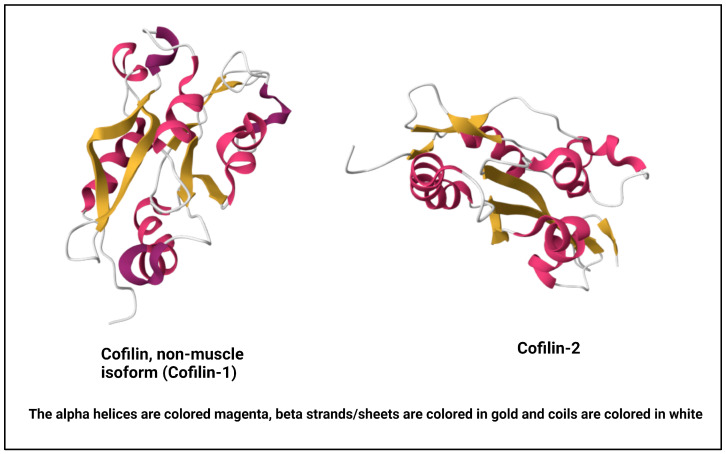


This activity is central to regulating actin dynamics within the cell [26]. The C-terminal region of cofilin contains phosphorylation sites critical for its regulation. The activity of cofilin is tightly regulated through a balance of phosphorylation and dephosphorylation events [27]. These regulatory mechanisms are crucial in modulating cofilin’s effects on the actin cytoskeleton. Serine-3 (Ser-3) is the primary phosphorylation site, and its phosphorylation status dictates cofilin’s activity [28]. LIM kinases (LIMKs) phosphorylate cofilin at Ser-3, leading to its inactivation [27,28,29]. This phosphorylation prevents cofilin from binding to F-actin, thereby inhibiting its ability to promote actin filament depolymerization [30]. Slingshot phosphatases, such as SSH1 and SSH2, dephosphorylate cofilin at Ser-3, restoring its activity. This dephosphorylation allows cofilin to bind to actin filaments, sever them, and enhance actin dynamics [31]. Apart from phosphorylation/dephosphorylation, cofilin’s activity is also pH-sensitive. It exhibits increased actin-binding and severing activity at lower pH levels, particularly relevant in cellular compartments with varying pH, such as synaptic terminals [32,33,34]. Moreover, other actin-binding proteins, such as profilin and tropomyosin, can modulate cofilin’s interaction with actin filaments, further fine-tuning its effects on cytoskeletal dynamics [35,36,37,38]. Understanding cofilin’s molecular structure and regulation is pivotal in deciphering its diverse cellular functions, including its involvement in neuronal processes, cell migration, and synaptic plasticity [39]. Dysregulation of these regulatory mechanisms can have profound implications for cellular function and has been linked to pathological conditions, including neurodegenerative diseases [40,41,42].

### 2.2. Role of Cofilin in Different Cellular Processes

Cofilin, a crucial regulator of actin cytoskeletal dynamics, plays a multifaceted role in various physiological processes, encompassing cell cycle regulation, cell motility, cell structure maintenance, autophagy, and apoptosis [13]. Cofilin helps in cytokinesis during cell division by contributing to contractile ring formation and dissolution, ensuring proper separation of daughter cells [43]. At the forefront of cell migration, cofilin actively promotes lamellipodia formation, facilitating the assembly and extension of thin, sheet-like protrusions rich in actin filaments. By severing and depolymerizing actin filaments, it contributes to dynamic lamellipodia remodeling, enabling cell movement and migration efficiency [44]). During cell migration, cofilin helps in lamellipodia formation and contributes to focal adhesions, which anchor the cell to the extracellular matrix and must undergo dynamic turnover. Cofilin contributes to this process by modulating actin dynamics at focal adhesion sites. It helps disassemble focal adhesions at the cell’s rear, allowing cells to detach and move forward [27]. Cofilin’s role in the motility of growth cones is particularly crucial in the context of neuronal systems, as growth cones are specialized structures at the tips of developing neurites (axon and dendrite growth processes) that play a fundamental role in guiding and navigating axons during neural development [45]. Cofilin’s involvement in growth cone motility is tightly linked to its ability to regulate the dynamic rearrangements of the actin cytoskeleton, a process essential for neurite outgrowth and guidance. The growth cone’s motility relies on the dynamic assembly and disassembly of actin filaments [46]. Actin filaments at the leading edge of the growth cone undergo polymerization, pushing the growth cone forward. At the same time, actin filaments at the rear of the growth cone undergo depolymerization, allowing forward movement [47]. By enhancing actin filament turnover, cofilin increases the pool of available actin monomers for filament assembly, promoting dynamic actin remodeling within the growth cone. Furthermore, cofilin’s activity is spatially and temporally regulated within the growth cone. At the leading edge, where active neurite outgrowth occurs, cofilin is involved in promoting actin dynamics. In contrast, cofilin may contribute to actin depolymerization at the central and rear regions, facilitating neurite retraction or turning. Growth cones navigate through chemotactic responses, where guidance cues direct their movement [48]. Cofilin-mediated actin dynamics are integral to the growth cone’s ability to interpret and respond to these guidance cues, allowing for precise axon pathfinding. Understanding the detailed molecular mechanisms of cofilin in growth cone motility provides insights into the fundamental processes that underlie neuronal development [49]. Dysregulation of cofilin activity in this context can have profound implications for axon guidance, neurite outgrowth, and the establishment of neural circuits, potentially contributing to neurodevelopmental disorders or compromised neuronal regeneration in the adult nervous system [50]. Cofilin also participates in intracellular trafficking by influencing the movement of vesicles, organelles, and other cargo within the cell, thus regulating the actin cytoskeleton’s organization [9]. This role of cofilin extends to the dynamic regulation of AMPA receptors. AMPA receptors are crucial for synaptic transmission and plasticity, and their proper trafficking is essential for maintaining synaptic function [51]. AMPA receptors mediate the majority of fast excitatory neurotransmissions in the central nervous system. Proper trafficking of AMPA receptors to and from the synapse is essential for synaptic plasticity, a process crucial for learning and memory [52]. Actin dynamics, regulated by proteins like cofilin, play a pivotal role in orchestrating these trafficking events. Dysregulation of actin-dependent AMPA receptor trafficking can contribute to excitotoxicity, a phenomenon where excessive activation of glutamate receptors leads to neuronal damage and death. This process is implicated in several pathological conditions, including AD and stroke [8,41]. In AD, aberrant AMPA receptor trafficking and excitotoxicity have been associated with synaptic dysfunction and neurodegeneration. Similarly, in stroke, the disruption of normal cellular processes, including AMPA receptor trafficking, contributes to excitotoxic neuronal damage following ischemic events [53].

Moreover, cofilin is vital for immune cell function, including chemotaxis, phagocytosis, and immune synapse formation. It enables immune cells to respond effectively to pathogens and foreign substances [54]. For an efficient chemotaxis effect, cofilin helps promote the flow of cytoplasmic material toward the cell’s leading edge, facilitating directed cell movement [55]. Particularly in immune cell migration and metastatic cancer cell invasion, cofilin’s actin cytoskeleton remodeling ability allows the cells to squeeze through tight spaces, a process known as amoeboid or bleb-based motility [30]. In neurons, it is involved in synaptic plasticity (a process essential for learning and memory) by impacting the remodeling of dendritic spines and, consequently, synaptic strength and morphology [39,56,57]. Dysregulation of cofilin can affect various cellular processes and has been implicated in pathological conditions, making it a significant focus of research in the context of neurodegenerative diseases and stroke. The following sections focus on the link between cofilin dysregulation and the pathophysiology of various neurodegenerative diseases and stroke.

## 3. Cofilin in Neurodegeneration

As previously mentioned, initially recognized for its pivotal role in cytoskeletal dynamics and cell motility, cofilin has emerged as a critical factor in developing neurodegenerative diseases [9]. Multiple lines of evidence connect cofilin dysregulation to these conditions, highlighting its contributions to neuronal dysfunction and degeneration. In AD, two major pathological features, amyloid-β (Aβ) and hyperphosphorylated tau, have been implicated in cofilin dysregulation. Aβ promotes cofilin dephosphorylation and activation, leading to excessive actin depolymerization, dendritic spine loss, synaptic dysfunction, and cognitive decline [5,41]. Hyperphosphorylated tau can also activate cofilin, contributing to the formation of neurofibrillary tangles and disruption of neuronal cytoskeletal structure [58]. In PD, α-synuclein, a key player in the disease’s pathogenesis, interacts with cofilin, potentially influencing actin dynamics and synaptic dysfunction [59]. Additionally, mitochondrial dysfunction in PD activates cofilin, leading to mitochondrial fragmentation and neuronal damage [60]. HD, characterized by the accumulation of mutant huntingtin protein, interacts with and activates cofilin. This interaction results in abnormal actin dynamics, neuronal cytoskeletal abnormalities, and synaptic dysfunction in HD-affected neurons [61]. Another neurodegenerative disease, amyotrophic lateral sclerosis (ALS), associated with TDP-43 protein pathology, affects cofilin phosphorylation and activity, contributing to cytoskeletal abnormalities and motor neuron degeneration [62].

In ischemic stroke, widespread neuronal injury and neuroinflammation is observed. Cofilin activation is implicated in ischemia-induced neuronal death and dendritic spine loss, adding to the neurological damage [9,63]. In the case of traumatic brain injury (TBI), axonal injury is a hallmark feature. Cofilin activation leads to axonal damage by promoting actin filament severing and destabilization [64,65]. Cofilin dysregulation has also been explored in various other neurodegenerative conditions, including frontotemporal dementia, multiple sclerosis, and prion diseases, suggesting its potential as a common pathological mechanism [4,66,67,68,69].

The evidence linking cofilin to neurodegenerative diseases underscores its role in mediating cytoskeletal dysfunction, synaptic impairment, and neuronal degeneration. Dysregulation of cofilin activity, often driven by alterations in its phosphorylation status, disrupts the delicate balance of actin dynamics in neurons, leading to dendritic spine loss, axonal damage, and impaired synaptic plasticity. This disruption plays a pivotal role in the pathophysiology of neurodegenerative diseases. Understanding the molecular mechanisms underlying cofilin dysregulation in these conditions is an active area of research, holding promise for developing targeted therapies aimed at preserving neuronal integrity and function in the face of neurodegeneration.

## 4. Cofilin Dysregulation in Stroke and Other Neurodegenerative Disorders

Ischemic stroke is a devastating neurological condition characterized by the sudden interruption of blood flow to the brain, resulting in the oxygen and nutrient deprivation of neurons [70]. Cofilin plays a significant role in the pathogenesis of stroke and other neurodegenerative diseases [71]. Here, we provide an overview of cofilin’s involvement in stroke and other neurodegenerative diseases and its implications for neuronal damage and neuroinflammation (Figure 2).

### 4.1. Ischemic Cascade

In response to ischemia, various cellular pathways are activated, leading to neuronal injury and death. Cofilin is a critical participant in these cascades [72]. Hypoxia and energy depletion following ischemia can lead to the dephosphorylation and activation of cofilin, promoting actin depolymerization [73].

### 4.2. Cytoskeletal Disruption

Activated cofilin can disrupt the actin cytoskeleton in neurons [40]. The activation of cofilin induces cytoskeletal disruption primarily through its ability to modulate actin dynamics. When cofilin is activated, it binds to actin filaments, which are the structural components of the cytoskeleton. Cofilin exhibits a preference for ADP-actin subunits within these filaments. Cofilin induces a conformational change in actin filaments, promoting the severing of these filaments into smaller fragments [74]. This process increases the number of free barbed ends essential for actin polymerization. Activated cofilin enhances the depolymerization of actin filaments by increasing the rate at which actin monomers dissociate from the filament ends [75]. This results in a higher turnover of actin filaments within the cytoskeleton. The increased depolymerization leads to a surplus of free globular (G)-actin monomers within the cellular pool, making them available for new filament assembly [76]. The combined actions of cofilin result in a shift in the dynamic equilibrium of actin filaments. There is an overall increase in the turnover of actin filaments, leading to a more dynamic and less stable actin cytoskeleton [77].

Loss of Cytoskeletal Integrity: The continuous severing and depolymerization induced by activated cofilin disrupt the structural integrity of the actin cytoskeleton. This dynamic instability can affect various cellular structures, including dendritic spines, synapses, and overall neuronal morphology. Cytoskeletal disruption contributes to neuronal damage and impairs neuronal functions [78].

### 4.3. Dendritic Spine Loss

Dysregulated cofilin activity, marked by its activation and subsequent influence on actin dynamics, plays a pivotal role in disrupting dendritic spine morphology and function [39]. Under normal conditions, cofilin regulates actin filament turnover, maintaining the structural plasticity of dendritic spines. However, when activated, cofilin enhances actin severing and depolymerization, destabilizing the actin cytoskeleton within spines [5]. This disruption extends to critical actin-dependent processes, including endocytosis and receptor recycling. Cofilin-induced alterations in actin dynamics impact the formation of endocytic vesicles, influencing the internalization of membrane components and synaptic receptors [79]. Concurrently, the compromised actin structure leads to abnormal dendritic spine morphology, characterized by filopodial protrusions and enlarged heads. Weakened actin filaments render dendritic spines susceptible to structural collapse, resulting in spine loss. This synaptic disconnection, attributed to dysregulated cofilin activity, contributes to impaired synaptic transmission and holds implications for neurodegenerative conditions, such as AD [5,41,80]. Ischemia-induced cofilin activation has been linked to dendritic spine loss in affected neurons [8]. This spine loss compromises synaptic connectivity and contributes to cognitive deficits observed in stroke patients [81].

### 4.4. Axonal Damage

Cofilin activation can promote the severing of actin filaments in axons, leading to axonal fragmentation and damage [82]. Axonal pathology, characterized by axonal swellings and spheroids formation, is a common feature in conditions like TBI and ALS [83]. Axonal damage disrupts neuronal communication and can lead to functional deficits [84]. Cofilin activation can lead to axonal damage in response to ischemia [8]. The severing of actin filaments in axons contributes to axonal fragmentation and impaired neuronal communication [85]. Axonal damage may further exacerbate neurological deficits and impair functional recovery post-stroke [72].

### 4.5. Neuronal Apoptosis

Dysregulated cofilin activity can trigger apoptotic pathways in neurons [86]. The cytoskeletal instability induced by cofilin activation can damage DNA and activate pro-apoptotic proteins [87]. Neuronal apoptosis contributes to losing viable neurons in the ischemic penumbra and core regions of the stroke-affected brain [88].

### 4.6. Neuroinflammation in Stroke, Its Implications and Cofilin as a Mediator of Neuroinflammatory Responses

Neuroinflammation plays a dual role in stroke, contributing to both protective and detrimental effects [89]. Chronic neuroinflammation is a hallmark of several neurodegenerative diseases and can exacerbate neuronal damage by activating immune cells and inducing oxidative stress [90,91,92]. Ischemic stroke triggers an immune response in the brain, activating immune cells such as microglia and attracting infiltrating leukocytes, including neutrophils and monocytes. Cofilin plays a crucial role in activating immune cells in the brain, influencing their migration, phagocytosis, and antigen presentation. This activation is prompted by the release of damage-associated molecular patterns (DAMPs) from injured neurons and astrocytes [93,94]. These activated immune cells release pro-inflammatory mediators, including cytokines like tumor necrosis factor-alpha (TNF-α) and interleukin-1β, chemokines, and reactive oxygen species (ROS) [95,96]. These inflammatory signals contribute to the recruitment of more immune cells and exacerbate neuronal damage [97,98]. It also impacts actin dynamics in immune cells, contributing to processes such as immune synapse formation and phagocytosis [99]. Cofilin activation in astrocytes can also contribute to neuroinflammation by promoting the release of pro-inflammatory cytokines and chemokines [90]. Additionally, cofilin indirectly influences blood–brain barrier (BBB) integrity, facilitating immune cell infiltration into the brain. Neuroinflammation can also disrupt the BBB, increasing its permeability. This allows immune cells to more easily infiltrate the brain and permits potentially harmful molecules, including pathogens and toxins, to enter [63,100]. Moreover, Cofilin’s activity is involved in the dynamic regulation of tight junctions between endothelial cells. Tight junctions are critical for forming a barrier that restricts the passage of substances between blood and brain tissue [101]. Cofilin’s ability to promote actin filament turnover influences the assembly and disassembly of tight junctions, thereby contributing to the permeability characteristics of the BBB. Cofilin-mediated actin dynamics are essential for the cytoskeletal remodeling of endothelial cells. This remodeling is necessary for processes such as cell migration, which is crucial during BBB development and repair [101,102]. Furthermore, neuroinflammation contributes to secondary brain injury, which extends beyond the initial ischemic insult and can persist for hours to days after the stroke. This secondary injury phase is characterized by ongoing neuronal death, edema, and an expansion of the infarct area [103,104,105]. Crucial roles are played by glial cells, including microglia and astrocytes, during neuroinflammation. Microglia transition to a pro-inflammatory state, producing cytokines and ROS, while reactive astrocytes release pro-inflammatory molecules, further propagating the inflammatory response [106,107,108]. In the neuroprotective role of inflammation during stroke, anti-inflammatory signals and immune cells are recruited to the injured area to dampen the inflammatory response, limiting further damage and promoting tissue repair and functional recovery [109,110,111]. The duration and intensity of neuroinflammation have significant implications for stroke recovery and rehabilitation outcomes. Excessive or prolonged inflammation may hinder recovery efforts, making it a challenging aspect of post-stroke care to balance the inflammatory response while promoting neuroprotection and neuroplasticity [12,112,113].

Studies have shown that the formation of cofilin-actin rods post-reperfusion underscores the dynamic response of neurons to ischemic insult, leading to disruptions in the cytoskeletal architecture and, consequently, impairments in organelle transport and dendritic spine loss. The increase in cofilin-1 levels, observed in the penumbra of the motor cortex following ischemia-reperfusion, implicates its potential role in gait imbalance and neuronal structural reorganization. The alterations in cofilin-1 mRNA and protein levels underline a complex regulatory mechanism during cerebral ischemia-reperfusion, potentially involving transcriptional and post-translational processes [96].

As previously mentioned, cofilin mediates neuroinflammatory responses in various neurological conditions, including stroke [8]. This involvement of cofilin in neuroinflammation highlights its potential as a therapeutic target for modulating neuroinflammatory processes. Understanding the role of cofilin in neuroinflammation is essential for exploring potential therapeutic strategies to modulate this complex process. Therefore, understanding the complexities of neuroinflammation has led to the development of various strategies, including immunomodulatory agents, anti-inflammatory drugs, and interventions to reduce BBB permeability. Emerging research explores anti-inflammatory approaches in combination with neuroprotective therapies [63,114,115,116]. A comprehensive understanding of the mechanisms involved in neuroinflammation is crucial for developing targeted therapies that harness its beneficial aspects while mitigating its detrimental effects, ultimately improving outcomes for stroke survivors.

## 5. Cofilin in Other Neurodegenerative Diseases

As mentioned earlier in this review, cofilin plays a significant role in regulating the actin cytoskeleton, cellular process and neuronal functions; thus, any disturbances in its structure/function have substantial consequences for various neurodegenerative diseases. Multiple studies suggested that cofilin dysregulation may have implications beyond stroke and could be involved in the pathological mechanisms of other neurodegenerative conditions such as AD, PD, schizophrenia, ALS and HD (Figure 3) [4,82,117].

### 5.1. Alzheimer’s Disease (AD)

AD is the major cause of dementia and is characterized by pathological hallmarks, including the accumulation of Aβ plaques and hyperphosphorylation tau-containing neurofibrillary tangles in the brain [118]. Cofilin dysregulation has been associated with the formation of these pathological hallmarks. Increased cofilin expression levels and actin remodeling contribute to synaptic dysfunction, dendritic spine loss, impaired neuronal plasticity, mitochondrial dysfunction, apoptosis, and neurotoxicity, ultimately leading to memory and cognitive impairment [41]. Moreover, the upregulation of cofilin-actin rods is observed in the brains of AD patients and AD mice models [119]. Rods develop within neurites, impairing synaptic function by dysregulation of cofilin activity, disturbing the typical dynamics of actin, inhibiting cellular transport, and aggravating the loss of mitochondrial membrane potential. The formation of rods is induced by the aggregation of Aβ and inflammatory cytokines that activate NADPH oxidase and generate reactive oxygen species [120]. Cofilin competes with tau for direct binding to microtubules by inhibiting tau-induced microtubule assembly in in-vitro models [121]. It has also been reported that genetic reduction of cofilin diminished tauopathy, mitigated synaptic dysfunction in tau-P301S mice, and reduced movement impairment in transgenic *C. elegans* [121]. Activated cofilin mediates neurotoxicity and contributes to the development of tauopathy by selectively interacting with tubulin, destabilizing microtubules and promoting tauopathy [121]. In addition, a study elucidated the mechanism underlying the transmission of tau by suggesting that the actin-binding protein cofilin undergoes cleavage by a cysteine protease called asparagine endopeptidase (AEP) at the N138 site in the brain of patients with AD. AEP-produced cofilin 1–138 fragments interact with tau and stimulate its aggregation. Overexpression of cofilin 1–138 in the brain of tau-P301S mice enables the exacerbation of pathological tau aggregates and contributes to cognitive deficit. These results shed light on the role of cofilin 1–138 in the aggregation and spread of tau, potentially implicating it in the development and progression of AD [58].

### 5.2. Parkinson’s Disease (PD)

PD is a progressive neurodegenerative condition characterized by motor dysfunction and loss of dopaminergic neurons in the substantia nigra of the brain, which leads to motor dysfunctions such as tremors, bradykinesia, dyskinesia, and rigidity [122]. The primary mechanism is represented by the aggregation of Lewy neurites and Lewy bodies in cortical neurons, leading to the misfolded α-synuclein in the presynaptic nerve terminal in the CNS [123,124]. The presence of extracellular α-synucleins leads to upregulation in the expression of glucose-related protein of 78 kDa (GRP78), an endoplasmic reticulum (ER) chaperone, on the neuronal plasma membrane, forming clusters in specific regions of the membrane. The interaction between α-synucleins and GRP78 initiates an ER signaling cascade that can trigger the phosphorylation/inactivation of cofilin. As a result, it promotes altering actin turnover and forming stress fibers, suggesting an early defect in synaptic function [125]. In addition, α-synuclein develops intracellular aggregates in the form of fibrils, which can be transmitted from cells that produce aggregates to cells without aggregates, ultimately leading to neuronal damage and the increased progression of the disorder. Cofilin interacts with α-synuclein to form combined fibrils that display a denser assembly and increase α-synuclein aggregation compared to pure α-synuclein fibrils, triggering neurotoxicity and impairment of mitochondrial and axonal function, suggesting that cofilin plays an essential role as a mediator in the aggregation and spread of α-synuclein in mice and humans [126]. The combined fibrils (cofilin and α-synuclein) exhibited higher pathogenicity and propagation of Parkinson’s features in mice than those composed solely of α-synuclein. This demonstrated that cofilin overexpression augmented the initiation, spread, and seeding of α-synuclein aggregates, leading to the development of Parkinson’s-like behavioral deficit in mice and neurotoxicity [59]. These findings highlighted the critical role of cofilin in the mediated pathogenic role and transmission of α-synuclein in the early stage and later progression of PD.

### 5.3. Schizophrenia

Schizophrenia represents a multifaceted neuropsychiatric disorder with a wide range of symptoms, such as hallucinations, delusion, loss of emotional expression, and cognitive deficits that impact memory function and alter synaptic plasticity [127,128]. In schizophrenia, a protein named 14-3-3 indirectly regulates the level of phosphorylated cofilin via δ-catenin signaling [129]. A study indicated that inhibition of 14-3-3 protein in neurons in the mouse results in behavioral impairment that resembles the symptoms of schizophrenia. The mutant mice (14-3-3 functional knockout mice) reduced phosphorylated cofilin and NMDA function, suggesting that disrupting 14-3-3 protein may lead to schizophrenia symptoms by interfering with actin dynamics.

### 5.4. Amyotrophic Lateral Sclerosis (ALS)

ALS is a progressive chronic motor neuron disease, while frontotemporal dementia (FTD) is associated with the degradation of neurons in the frontal and temporal lobes [130]. Studies have exhibited dysregulation of cofilin expression activity in ALS and FTD models, which may suggest its involvement in altering actin dynamics, neurotoxicity, and synaptic deficit [9,62]. C9ORF72, a frequently occurring genetic factor in ALS and FTD, alters the activity of the small GTPase Arf6, leading to increased LIMK action and cofilin phosphorylation that can reduce axonal actin dynamics and initiate neuronal loss (neurotoxicity). Upregulation of phosphorylated cofilin in C9ORF72-depleted motor neurons has been observed in lymphoblastoid cells derived from patients and post-mortem brain samples with ALS. This upregulation can be reversed by dominant negative Arf6, suggesting that C9ORF72 regulates small GTPases, influencing actin dynamics in axons [131]. Moreover, TAR-binding protein 43 (TBP-43), misfolded and mislocalized, is reported in the motor neurons of most ALS patients. Dysregulation of actin dynamics is associated with the overexpression of phosphorylated LIMK and phosphorylated cofilin in patients with sporadic ALS.

### 5.5. Huntington’s Disease (HD)

Another neurodegenerative disease where cofilin has gained attention is HD [61]. It is an inherited disorder (mutation of the Huntingtin gene, HTT) associated with the deterioration of cortical striatal pathways and abnormal brain structure and metabolism, leading to motor and cognitive impairment [132,133]. Cofilin dysregulation impairs microglial cell migration and abolishes the dynamic and extension of microglia processes in the cortex of HD mice, suggesting a reduction in the expression level of cofilin that regulates actin cytoskeleton dynamics [134]. Cofilin is critical in connecting structural alteration in dendritic spines with synaptic function by regulating the trafficking of AMPA receptors (AMPAR). The temporary activation of cofilin through dephosphorylation facilitates the insertion of AMPAR at the synapse terminal, and the inactivation of cofilin via LIMK-mediated cofilin phosphorylation allows the enlargement of dendritic spines, enabling the dynamic regulation of synaptic plasticity and spine morphology in HD, AD, and stroke [51]. In the HTT-depleted neuron model, spine shape is altered due to the diminished activity of AMPAR-mediated spine currents, consistent with findings from a previous study that showed the decline of wild-type HTT in cortical neuron progenitors produces abnormal synapse number and altered dendritic spine morphology and negatively impacts its function in mice [135,136]. Furthermore, the absence of cofilin activation, particularly in the context of HTT loss, leads to the hyperactivation of LIMK and stabilization of the actin, which promotes the formation of large dendritic spines and inhibits the recruitment of AMPA receptors (AMPAR) to synapses in response to synaptic activity [61]. Thus, this may explain the decrease in AMPAR expression in the postsynaptic region of HTT-depleted neurons, which may contribute to the eventual progression of HD-related symptoms.

Collectively, cofilin dysregulation has implications beyond stroke, suggesting its involvement in other neurodegenerative diseases. Understanding the role of cofilin in the studies mentioned above may provide novel insights into cofilin signaling involving pathological pathways and potential therapeutic interventions. Continued research and developmental efforts are needed to elucidate the specific mechanisms by which cofilin contributes to these diseases, ultimately paving the way for novel and targeted treatments for neurodegenerative diseases and stroke.

## 6. Cross-Disciplinary Insights into Targeting Cofilin for Neuroprotection

Targeting cofilin for neuroprotection and attenuating neurotoxicity has promising potential in stroke and neurodegenerative diseases [11,82]. Researchers have employed multidimensional approaches to explore the cross-disciplinary insights into targeting cofilin for neuroprotection, leveraging knowledge from miscellaneous fields such as neuroscience, cellular and molecular biology, and bioinformatics. Modulating the expression of cofilin or its signaling cascades, such as the Rho family, SSH1, and LIMK, leads to restoring the normal function of cofilin [73]. This strategy underlying the potential therapeutic intervention of cofilin showed substantial results in preclinical stroke and neurodegenerative disease models [71,95,137,138]. Several studies have investigated the role of cofilin in neuronal survival and synaptic plasticity [139,140,141]. To explore the intricacy of cofilin and synaptic function, researchers utilized imaging and electrophysiology techniques that reveal the role of cofilin in preserving synaptic structure [121,142]. In addition, computational modeling and bioinformatics are vital for understanding cofilin dynamics and identifying potential drug targets [143,144]. This can be accomplished by exploring large-scale data and analyzing gene expression patterns, protein interaction, and transitional modification, which contribute to cofilin dysregulation and facilitate the design of potential target intervention [145]. Collaborative integration between scientists from several disciplines, including neuroscientists, molecular biologists, and computational biologists, is critical for developing and further advancing the therapeutics field of targeting cofilin.

## 7. Therapeutic Strategies, Including Cofilin Inhibitors

Various potential therapeutic strategies are under investigation to modulate cofilin expression or its signaling cascades in neurodegenerative diseases. These approaches seek to inhibit cofilin to mitigate the pathological pathways associated with these conditions. One potential therapeutic approach involves inhibiting cofilin. In neurodegenerative diseases, dysregulation of cofilin or the formation of cofilin–actin rods may contribute to neuronal damage, degeneration, and further altered cellular pathways. Therefore, inhibiting cofilin could prevent or attenuate its pathological effects [4,9,73,82]. This can be achieved by modulating cofilin phosphorylation or kinase-regulated cofilin expression, utilizing gene therapy and developing a small molecule targeting cofilin or its signaling pathways.

Cofilin activity is regulated by kinases such as LIMK, which phosphorylates and inactivates cofilin, and slingshot-1 (SSH1)/calcineurin, which reactivates cofilin. Modulation of cofilin phosphorylation or kinases like LIMK can indirectly impact cofilin [146]. A study conducted by Deng et al. showed that the administration of a peptide that inhibits the dephosphorylation of cofilin in familial AD (FAD) mice resulted in a partial improvement in the surface expression of AMPA and NMDA receptor subunits, restoration of synaptic currents mediated by AMPAR and NMDAR, and improvements in working memory and novel object recognition. These findings indicate that targeting the cofilin–actin signaling pathway shows promise in altering the physiological and behavioral abnormalities associated with AD [147]. In addition, a study showed that injected Adeno-associated viruses (AAVs) encoded with asparagine endopeptidase (AEP)-induced cofilin 1–138 mutant led to a reduction in pathological tau and cognitive deficit compared with the control group (injected with AVVs encoded with wild-type cofilin) in a mouse model of AD (tau-P301S) [58]. Furthermore, a study demonstrated that introducing RFP-LIMK1 into the cortical penumbra using lentivirus 7 days prior to transient middle cerebral artery occlusion (tMCAO) rat model significantly mitigated cofilin rods, MAP2 degradation, and ischemia-mediated apoptosis compared to the rats injected with RFP control 24 h after tMCAO, suggesting the participation of cofilin in the initiation of neuronal cell death after ischemia and potential therapeutic intervention in ischemic stroke [7].

Several studies are investigating the potential of gene therapy, such as small interfering RNA (siRNA), to modulate cofilin expression. Decreasing the expression of cofilin at the genetic level aims to restore its normal function and alleviate the neurodegenerative processes associated with various disorders [137]. Our group conducted numerous studies utilizing small interfering RNA (siRNA) targeting cofilin to mitigate its detrimental impact. Treatment of cortical neurons subjected to OGD with cofilin siRNA significantly reduced the cofilin expression and caspase-3 cleavage and improved cell survival compared with the scramble siRNA (control group) [148]. Inhibiting cofilin using cofilin siRNA in LPS-induced microglia activation led to decreasing the proliferation and migration of microglia and attenuating the inflammatory mediators such as inducible nitric oxide synthase (iNOS), nitric oxide (NO), tumor necrosis factor-α TNF-α, and COX2 by modulating the nuclear factor κB (NF-κB) and JAK–STAT pathways [95]. Another study demonstrated that challenging microglia with hemin increased cofilin and NO expression; however, knockdown of cofilin using cofilin siRNA notably reduced oxidative stress (e.g., iNOS), ER stress (e.g., Wfs-1, XBP-1, and spliced XBP-1), and inflammation (e.g., TNF-α and interleukin-1β (IL-1β)) compared to scramble siRNA [149]. In the mouse model of intracerebral hemorrhage (ICH), inhibition of cofilin by cofilin siRNA before the experimental induction of the ICH showed a remarkable improvement of sensorimotor tests and attenuated cofilin expression, hematoma volume, brain edema, BBB impairment, microglia activation, apoptosis, and oxidative and nitrosative stress [150]. The aforementioned studies provided novel insights into gene therapy as a potential therapeutic strategy in modulating cofilin expression.

A novel approach to inhibiting cofilin implies the development of small molecule inhibitors specifically designed to target cofilin overexpression/dysregulation and its signaling. A small molecule could be developed to modulate cofilin signaling activity selectively and effectively, thereby suggesting a potential therapeutic strategy for various conditions associated with cofilin dysregulation. Our research group successfully developed a small molecule cofilin inhibitor directly targeting cofilin or its signaling regulators and the pathological cascades involved in neurodegenerative diseases mediated by cofilin dysregulation. By regulating cofilin, it might be possible to prevent aberrant actin dynamics, neuroinflammation, oxidative stress, apoptosis, synaptic dysfunction, and neurodegeneration [11,64,71]. A novel first-in-class small molecule inhibitor (CI or SZ3) targeting cofilin, developed by our research group, represents a remarkable advancement in addressing neuroinflammation [71]. The developed CI significantly decreased cofilin severing activity and microglia activation in the LPS in vitro model by attenuating microglia migration, proliferation, and inflammation. In addition, in the thrombin in vitro model simulating hemorrhagic stroke, the CI reduced inflammatory mediators such as TNF-α, NO, and PAR1. CI also increased cell viability in neurons by decreasing NF-κB, caspase-3, and HtrA2 [71]. Furthermore, attenuating the expression of SSH1 subsequently led to decreased cofilin overexpression using a CI in the in vitro H_2_O_2_ and in vivo TBI model [64]. The CI decreased microglial activation and inflammatory cascades and provided protection against reactive oxygen species and apoptosis in neurons. In the TBI mouse model, the CI-activated Nrf_2_ diminished oxidative and nitrosative markers such as iNOS, NOX4, NOX2, and 3-nitrotyrosine (3NT) [64]. A study evaluating the metabolic stability, pharmacokinetics, and dose-proportionality of this CI by LC-MS/MS revealed promising pharmacological and pharmacokinetic properties at 10 mg/kg, suggesting that this CI could be a viable treatment option for neurodegenerative diseases [151]. Taken together, our group showed a significant role of the novel CI as a potential therapeutic strategy in various neurodegenerative diseases.

Understanding that these potential therapeutic approaches targeting cofilin showed promising results in the preclinical stage is essential. However, further research is needed to pave the way to the clinical setting and evaluate CI efficacy, safety, and feasibility in humans to counter pathological processes in neurodegenerative diseases. Various cofilin inhibitors studied in different disease models are listed in Table 1.

## 8. Challenges and Future Directions

Developing cofilin inhibitors that selectively target its pathological activity without interfering with essential physiological functions remains a challenge, as is the case with other inhibitor classes of drugs. Achieving a delicate balance between inhibition and preservation of necessary cellular processes is crucial [152]. Cofilin inhibitors may inadvertently affect other cellular processes or proteins with similar structural features or have off-target effects. Ensuring the specificity of inhibitors is essential to avoid unintended consequences [153]. To achieve this, multidimensional approaches could be incorporated, such as Structure–Activity Relationship studies, computational approaches, selectivity screening, chemoproteomics and affinity profiling, in silico predictions, and negative control compounds, among others. Moreover, assessing the long-term safety and tolerability of cofilin inhibitors, particularly in chronic neurodegenerative conditions, is essential. Potential side effects or immunogenicity must be thoroughly investigated [154]. Moving from preclinical research to clinical trials presents regulatory and logistical challenges. Demonstrating the safety and efficacy of CI in humans is a complex and resource-intensive process.

Future research in the field of cofilin signaling and its role in neurodegenerative diseases holds promise in several areas. Biomarker development to identify reliable markers associated with cofilin dysregulation can aid in patient stratification and treatment monitoring. Exploring combination therapies that target cofilin in conjunction with other relevant pathways, such as neuroinflammation and oxidative stress, may yield synergistic neuroprotective effects. Precision medicine can optimize treatment outcomes by tailoring cofilin-targeted therapies to individual patient-based disease characteristics and genetic profiles. Innovative drug delivery systems, such as nanoparticles or targeted biologics, can enhance the efficiency of cofilin inhibitor delivery to specific brain regions. Leveraging synthetic biology approaches to design novel molecules with enhanced specificity and potency as cofilin inhibitors may lead to more effective therapeutic agents.

## 9. Conclusions

Cofilin’s multifaceted role in neurodegeneration and stroke has become apparent. It extends beyond its traditional function, contributing to neuroinflammation, cytoskeletal disruption, neuronal apoptosis, and mitochondrial dysfunction. Cofilin holds promise as a therapeutic target for various neurodegenerative diseases. Tailored therapies based on disease and cofilin involvement, innovative drug delivery, and precision medicine offer potential neuroprotection. A call to action for collaborative research emphasizes the importance of identifying biomarkers and advancing translational efforts. This ongoing journey requires collective effort and unwavering commitment to improving neuroprotection and patient care in the face of debilitating neurological conditions.

## Figures and Tables

**Figure 2 cells-13-00188-f002:**
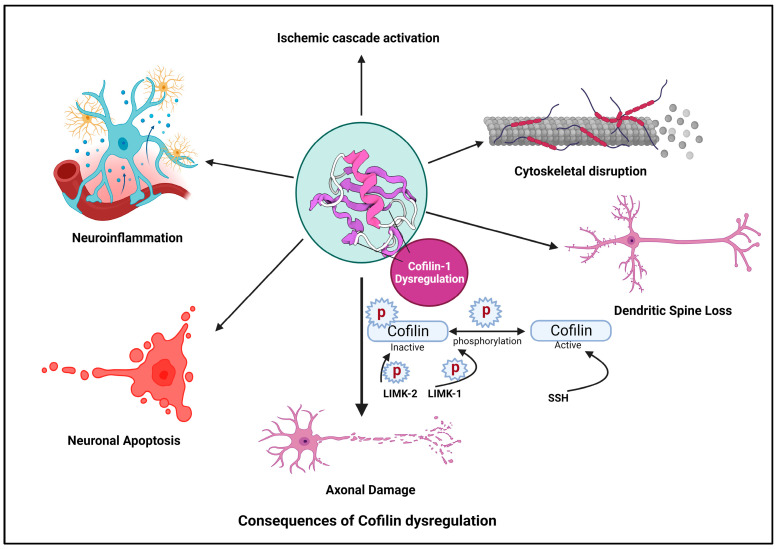
Cofilin dysregulation leads to neuroinflammation and neuronal apoptosis.

**Figure 3 cells-13-00188-f003:**
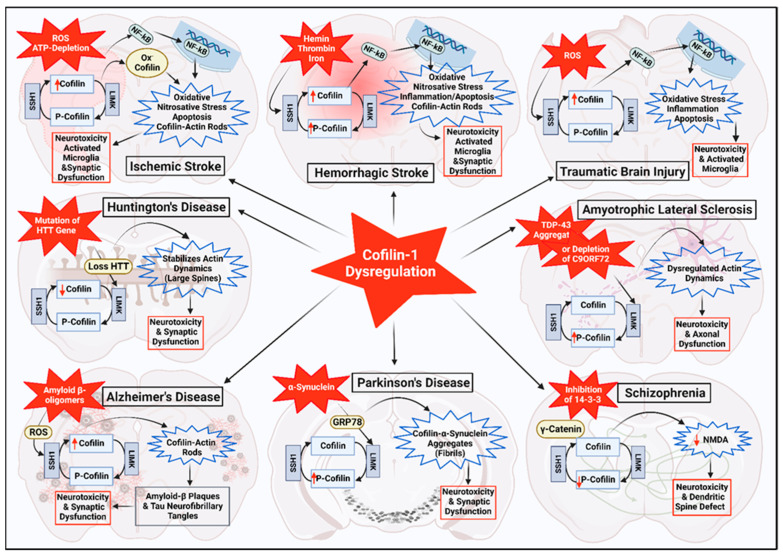
Dysregulation of cofilin in neurologic and neurodegenerative diseases. Red upward arrows indicate an increase and red downward arrows indicate a decrease.

**Table 1 cells-13-00188-t001:** Therapeutic interventions and approaches employing cofilin inhibition.

Cofilin Inhibitor Strategy	Model	Tissue/Cell Specificity	Main Outcome	Study
Cofilin dephosphorylation inhibitory peptide	Familial Alzheimer’s disease (FAD) model	Frontal cortical neurons	- Improved AMPAR- and NMDAR-facilitated synaptic currents - Improved working memory	[147]
LIM kinase (Limk1) overexpression using RFP-LIMK lentivirus	Transient middle cerebral artery occlusion (tMCAO) in rats	Cortical penumbra (neuron)	- Reduced apoptosis - Reduced cofilin rod formation and MAP2 degradation	[7]
Adeno-associated viruses (AAVs) encoded with AEP induced cofilin 1–138 mutant	Tau-P301S, Alzheimer’s disease mouse model	Cortex (cortical primary neurons)	- Decreased synaptic dysfunction - Decreased cognitive impairments	[58]
Cofilin siRNA	Oxygen–glucose deprivation/reperfusion (OGD/R) in primary cortical neurons	Neuron	- Increased cell viability - Decreased apoptosis	[148]
Cofilin siRNA	Microglial cell line challenged with LPS	Microglia	- Reduced inflammatory mediators	[95]
Cofilin siRNA	Microglial cell line challenged with hemin	Microglia	- Decreased inflammation - Decreased oxidative stress - Decreased ER stress	[149]
Cofilin siRNA	Intracerebral hemorrhage model in mice	Around the hemorrhage	- Improved sensorimotor tests - Reduced hematoma volume, brain edema, and blood–brain barrier (BBB) disruption - Reduced microglial activation, apoptosis, and oxidative and nitrosative stress	[150]
Small molecule cofilin inhibitor (CI or SZ-3)	Microglia cell line challenged with LPS and thrombin Neuron cell line challenged with thrombin	Microglia and neuron	- Reduced neuroinflammation in microglia - Improved cell viability in neurons - Reduced neuroinflammation and apoptosis in neurons	[71]
Small molecule cofilin inhibitor (CI or SZ-3)	H2O2-induced oxidative stress model in microglia (HMC3) and neuron (SH-SY5Y) cell line and traumatic brain injury (TBI) in mice	Microglia and neuron as well as around the injury in the TBI model	- Decreased microglia activation and inflammatory mediators - Decreased ROS accumulation and apoptosis, as well as increased antioxidant mediators in neurons - Decreased the expression of oxidative/nitrosative stress markers in mice with TBI	[64]

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
