# Peer review of "The Multifaceted Role of Cofilin in Neurodegeneration and Stroke: Insights into Pathogenesis and Targeting as a Therapy"

_cells, 2024, doi:10.3390/cells13020188_

Round 1

Reviewer 1 Report

Comments and Suggestions for Authors

The review does not provide new information that other published reviews don't have. The authors mentioned that they will focus on the role of cofilin in stroke, but if we put the different paragraphs and lines about this together, they won't fill more than a page.    

Comments on the Quality of English Language

The review needs some language revision, grammar and sentence construction. Many of the sentences are repetitive.  

Author Response

The review does not provide new information that other published reviews don't have. The authors mentioned that they will focus on the role of cofilin in stroke, but if we put the different paragraphs and lines about this together, they won't fill more than a page.

Response: In the present review, we have focused more on the therapeutic approaches targeting cofilin-1. In this context, we discussed the potential mechanism of a novel first-in-class cofilin inhibitor (SZ3) from our lab. We believe it is an important addition to the literature review and distinguishes our manuscript from the other published reviews. We have modified the title of the manuscript accordingly.   

The review needs some language revision, grammar and sentence construction. Many of the sentences are repetitive.

Response: The issues have been addressed in the revised manuscript.

Reviewer 2 Report

Comments and Suggestions for Authors

The review on the role of cofilin in neurodegeneration, with a focus on insights into stroke pathogenesis is overall well written and a valuable contribution to the field. However, there are several aspects in which the manuscript could/should be improved:

The wording in the abstract suggests that 14-3-3 involvement / dysfunction is specific to Schizophrenia, which it is not. 14-3-3 is also involved e.g. in Parkinson's and Alzheimer's disease. An additional sentence prior to the sentence, starting 'Moreover...' in line 26, may allow to point out that there is likely overlap in the pathologies and involvement of cofilin in the different diseases. For example, excitotoxicity is shared by Alzheimer's disease and stroke. Reference to cofilin dysfunction could be put into the context of share pathologies in that regards.

In the abstract, line 28, challenges in inhibitor development should also focus on the challenges of achieving tissue / cell type specificity when targeting cofilin. This is also only superficially covered in the actual discussion in the review and should be carved out more. For example, what systems are available to target the manipulation of cofilin activity-modifying proteins to cells and tissue of interest (using different capsids of viral vector systems and/or neuron-specific / glia-specific promoters, etc).

The rationale for focusing on stroke in this review is not as well carved out. Why is stroke of particular interest, compared to the neurodegenerative diseases also discussed? Therefore, is the title really representative of the review focus?

The figures should include more detailed figure legends. For example, Figure 1 should have a detailed figure legend, outlining the significance of the colour coding, etc.

The comment on the importance of understanding cofilin's molecular structure (line 106) is good. However, a discussion on the molecular structure is largely lacking in this review. Maybe a paragraph / section on this aspect would be helpful.

The discussion on the role of cofilin in cell migration (line 120 and following) is helpful. However, it is unclear why more relevant reference to the detailed analysis of cofilin function in the motility of growth cones (more relevant to neuronal systems) is missing. What about growth cone regulation, which is a lot more relevant to neuronal cells. For example, Zhang et al., 2019, JCB. 

Lines 124 and 125: the role of cofilin in intracellular trafficking is mentioned at this stage. This would be the ideal place to introduce the importance of actin-dependent AMPA receptor trafficking. Dysfunction of this could contribute to excitotoxicity, as observed in several pathological conditions, including Alzheimer's disease and stroke, as mentioned above.

Lines 160-164: The recognition of potential common pathological mechanisms is good. However, making this conclusion from the previous sentence, merely stating that cofilin dysregulation has been observed in different diseases, is a bit of a stretch. This should be elaborated on further.

The section 4.2 on cytoskeletal disruption is a bit superficial for that it is key to this review.

Similarly, section 4.3 needs more detail, discussing for example actin-dependent processes such as endocytosis and receptor recycling in dendritic spines.

In figure 2, it would be helpful to include evidence for upstream mechanisms that lead to cofilin dysregulation.

Line 228: Are there 'only' indirect effects on the BBB? What about the role of actin in cells that contribute to the formation and maintenance of the BBB? This would be worth discussion at this stage.

Line 275: The findings in Woo et al. (2019) on cofilin competing with tau binding to microtubules are largely in vitro data. Therefore, the wording should be carefully adjusted, as a competition in vivo is less clear, where tau shows a high on- and off- rate in binding to microtubules (refer to kiss-and-hopp model, described by the Brandt lab). 

Lines 340 and following: The dysfunction of AMPAR trafficking is not specific to HD (see comments above).

Lines 386 and following: Outline here the challenges of targeting the inhibition to correct tissues / cells.

Table 1: include a column addressing tissue-specificity of targeting.

Line 461: 'Ensuring the specificity...' in this paragraph, it should be discussed what the approaches could be to ensure specificity.

Minor points / spelling errors:

Line 39: change 'represent a elusive' to 'represent an elusive'

Line 295: change 'a clustered' to 'clusters'

Line 299: change 'to unaggregated' to 'cells without aggregates'

Line 390: inconsistency in introducing the abbreviation LIMK (already used earlier in the text).

Line 393: Deng et al. is missing the year of reference.

Comments on the Quality of English Language

Overall, the review is very well written. There are some typos that should be corrected as outlined above.

Author Response

The review on the role of cofilin in neurodegeneration, with a focus on insights into stroke pathogenesis is overall well written and a valuable contribution to the field. However, there are several aspects in which the manuscript could/should be improved:

Response: We thank the reviewer for taking the time to review our manuscript thoroughly.

The wording in the abstract suggests that 14-3-3 involvement / dysfunction is specific to Schizophrenia, which it is not. 14-3-3 is also involved e.g. in Parkinson's and Alzheimer's disease. An additional sentence prior to the sentence, starting 'Moreover...' in line 26, may allow to point out that there is likely overlap in the pathologies and involvement of cofilin in the different diseases. For example, excitotoxicity is shared by Alzheimer's disease and stroke. Reference to cofilin dysfunction could be put into the context of share pathologies in that regards.

 Response: Thank you for pointing this out; we have mentioned 14-3-3 involvement in Alzheimer's and Parkinson's diseases as well. We also added a sentence before the word Moreover.

In the abstract, line 28, challenges in inhibitor development should also focus on the challenges of achieving tissue / cell type specificity when targeting cofilin. This is also only superficially covered in the actual discussion in the review and should be carved out more. For example, what systems are available to target the manipulation of cofilin activity-modifying proteins to cells and tissue of interest (using different capsids of viral vector systems and/or neuron-specific / glia-specific promoters, etc).

Response: We addressed the reviewer's note on achieving tissue/cell specificity. We reported the published work on this context in section 7. However, future research is warranted for utilizing different capsids of viral vector systems or employing neuron-specific and glia-specific promoters, which could provide precise control over the modulation of cofilin activity in desired cells and tissues. Targeted delivery systems are an example of investigating the effects of manipulating cofilin in specific cellular contexts, thereby enhancing our understanding of its role in neurodegenerative diseases and potentially opening new prospects for therapeutic interventions.

The rationale for focusing on stroke in this review is not as well carved out. Why is stroke of particular interest, compared to the neurodegenerative diseases also discussed? Therefore, is the title really representative of the review focus?

Response: We have modified the title of the revised manuscript.

The figures should include more detailed figure legends. For example, Figure 1 should have a detailed figure legend, outlining the significance of the colour coding, etc.

Response: Changed as per the reviewer suggestions

The comment on the importance of understanding cofilin's molecular structure (line 106) is good. However, a discussion on the molecular structure is largely lacking in this review. Maybe a paragraph / section on this aspect would be helpful.

Response: Modified as per reviewer suggestions.

The discussion on the role of cofilin in cell migration (line 120 and following) is helpful. However, it is unclear why more relevant reference to the detailed analysis of cofilin function in the motility of growth cones (more relevant to neuronal systems) is missing. What about growth cone regulation, which is a lot more relevant to neuronal cells. For example, Zhang et al., 2019, JCB.

Response: We incorporated a paragraph and explained the role of cofilin in the growth cone in section 2.2.

Lines 124 and 125: the role of cofilin in intracellular trafficking is mentioned at this stage. This would be the ideal place to introduce the importance of actin-dependent AMPA receptor trafficking. Dysfunction of this could contribute to excitotoxicity, as observed in several pathological conditions, including Alzheimer's disease and stroke, as mentioned above.

Response: Modified as suggested by reviewer.

Lines 160-164: The recognition of potential common pathological mechanisms is good. However, making this conclusion from the previous sentence, merely stating that cofilin dysregulation has been observed in different diseases, is a bit of a stretch. This should be elaborated on further.

Response: Modified as as suggested.

The section 4.2 on cytoskeletal disruption is a bit superficial for that it is key to this review.

Response: We have added more details on the cytoskeletal disruption in the revision.

Similarly, section 4.3 needs more detail, discussing for example actin-dependent processes such as endocytosis and receptor recycling in dendritic spines.

Response: Modified as suggested by the reviewer.

In figure 2, it would be helpful to include evidence for upstream mechanisms that lead to cofilin dysregulation.

Response: Modified as suggested by the reviewer.

Line 228: Are there 'only' indirect effects on the BBB? What about the role of actin in cells that contribute to the formation and maintenance of the BBB? This would be worth discussion at this stage.

Response: Thank you for pointing this out. We have added a paragraph in the revised manuscript to address this.

Line 275: The findings in Woo et al. (2019) on cofilin competing with tau binding to microtubules are largely in vitro data. Therefore, the wording should be carefully adjusted, as a competition in vivo is less clear, where tau shows a high on- and off- rate in binding to microtubules (refer to kiss-and-hopp model, described by the Brandt lab).

Response: This reviewer has raised an important point regarding the findings of Woo et al. (2019) and the need to carefully adjust the wording to reflect the in vitro nature of the data in the context of competing with tau binding. Thus, we corrected the sentence and clarified that it is in an in-vitro model.

Lines 340 and following: The dysfunction of AMPAR trafficking is not specific to HD (see comments above).

Response: We added Alzheimer's and stroke along with HD in this context.

Lines 386 and following: Outline here the challenges of targeting the inhibition to correct tissues / cells.

Response: We thank this reviewer for pointing it out; however, the challenges for targeting cofilin were already discussed in section 8.

Table 1: include a column addressing the tissue-specificity of targeting.

Response: In the revised manuscript, we have included this column in the table.

Line 461: 'Ensuring the specificity...' in this paragraph, it should be discussed what the approaches could be to ensure specificity.

Response: We have mentioned the various approaches in the revision.

Minor points / spelling errors:

Line 39: change 'represent a elusive' to 'represent an elusive'

Line 295: change 'a clustered' to 'clusters'

Line 299: change 'to unaggregated' to 'cells without aggregates'

Line 390: inconsistency in introducing the abbreviation LIMK (already used earlier in the text).

Line 393: Deng et al. is missing the year of reference.

Response: The above-mentioned errors were incorporated in the revision.

Reviewer 3 Report

Comments and Suggestions for Authors

The review is focused on the role of the cofilin signaling pathway in the course of stroke pathogenesis, thus paragraphs related to Parkinson's disease, Huntington's disease, and others should be removed. Also, the Table should be corrected and focused only on the paper subject. Eventually, the title should be changed and refer in general to neurodegenerative diseases.

All cited references should be numbered, please correct paragraph 3, page 4.

Page 9, paragraph 7 -.....including cofilin inhibitors (not Inhibitors)

Table 1 is not cited within the text.

Table - please, put the name of small molecule cofilin inhibitors cited in

Describe more the potential cofilin inhibitors, among them the most advanced seems to be SZ-3. Could you add more data concerning its current development in the context of stroke or neurodegeneration?

Comments on the Quality of English Language

Minor editing of English language required

Author Response

The review is focused on the role of the cofilin signaling pathway in the course of stroke pathogenesis, thus paragraphs related to Parkinson's disease, Huntington's disease, and others should be removed. Also, the Table should be corrected and focused only on the paper subject. Eventually, the title should be changed and refer in general to neurodegenerative diseases.

Response: We have modified the title in the revised manuscript.

All cited references should be numbered, please correct paragraph 3, page 4.

Response: Thank you for pointing this out. We have changed these accordingly.

Page 9, paragraph 7 -.....including cofilin inhibitors (not Inhibitors)

Response: Corrected in the revision.

Table 1 is not cited within the text.

 Response: We have included Table 1. In the text.

Table - please, put the name of small molecule cofilin inhibitors cited in

Describe more the potential cofilin inhibitors, among them the most advanced seems to be SZ-3. Could you add more data concerning its current development in the context of stroke or neurodegeneration?

Response: We have a paper under review that discusses the SZ-3 in stroke, such as intracerebral hemorrhage in mice. We also have studies that are still unpublished about SZ-3 in traumatic brain injury in mice and during chronic inflammation.